# Phloem-Mobile *MYB44* Negatively Regulates Expression of *PHOSPHATE TRANSPORTER 1* in Arabidopsis Roots

**DOI:** 10.3390/plants12203617

**Published:** 2023-10-19

**Authors:** Toluwase Olukayode, Jieyu Chen, Yang Zhao, Chuanhezi Quan, Leon V. Kochian, Byung-Kook Ham

**Affiliations:** 1Global Institute for Food Security (GIFS), University of Saskatchewan, 421 Downey Rd, Saskatoon, SK S7N 4L8, Canada; toluwase.olukayode@gifs.ca (T.O.); jieyu.chen@gifs.ca (J.C.); yang.zhao@gifs.ca (Y.Z.); chuanhezi.quan@gifs.ca (C.Q.); leon.kochian@gifs.ca (L.V.K.); 2Department of Biology, University of Saskatchewan, 112 Science Place, Saskatoon, SK S7N 5E2, Canada; 3Department of Plant Science, University of Saskatchewan, 51 Campus Drive, Saskatoon, SK S7N 5A8, Canada

**Keywords:** Pi-starvation stress, mobile mRNA, phloem, systemic signaling

## Abstract

Phosphorus (P) is an essential plant macronutrient; however, its availability is often limited in soils. Plants have evolved complex mechanisms for efficient phosphate (Pi) absorption, which are responsive to changes in external and internal Pi concentration, and orchestrated through local and systemic responses. To explore these systemic Pi responses, here we identified *AtMYB44* as a phloem-mobile mRNA, an Arabidopsis homolog of *Cucumis sativus MYB44*, that is responsive to the Pi-starvation stress. qRT-PCR assays revealed that *AtMYB44* was up-regulated and expressed in both shoot and root in response to Pi-starvation stress. The *atmyb44* mutant displayed higher shoot and root biomass compared to wild-type plants, under Pi-starvation conditions. Interestingly, the expression of *PHOSPHATE TRANSPORTER1;2* (*PHT1;2*) and *PHT1;4* was enhanced in *atmyb44* in response to a Pi-starvation treatment. A split-root assay showed that *AtMYB44* expression was systemically regulated under Pi-starvation conditions, and in *atmyb44*, systemic controls on *PHT1;2* and *PHT1;4* expression were moderately disrupted. Heterografting assays confirmed graft transmission of *AtMYB44* transcripts, and *PHT1;2* and *PHT1;4* expression was decreased in heterografted *atmyb44* rootstocks. Taken together, our findings support the hypothesis that mobile *AtMYB44* mRNA serves as a long-distance Pi response signal, which negatively regulates Pi transport and utilization in Arabidopsis.

## 1. Introduction

Phosphorus (P) is an essential element that is involved in various functionalities in living organisms, including plants. Therefore, it is critical to ensure proper phosphorus nutrition for optimal plant growth and development, by supplying phosphate (Pi) fertilizers to maximize crop productivity in modern agriculture systems [1,2]. However, soil-applied Pi fertilizer becomes rapidly immobilized, due to chemical reactions with cations (e.g., magnesium, calcium, aluminum, and iron) in both alkaline and acidic soils [3,4]. This can limit the availability of applied Pi to the plant to around 20–30% for plant use [3,4]. Furthermore, Pi is a finite and non-renewable resource; this has led to increased interest in understanding the mechanisms by which plants absorb and use Pi under limiting Pi conditions. Here, the goal is to develop elite plant lines with enhanced P acquisition and utilization efficiency traits [3,4,5,6,7].

Plants have evolved sophisticated adaptative mechanisms, which involve a range of physiological, morphological, biochemical, and molecular processes, to respond to low Pi conditions for efficient Pi foraging in soils [4,5,8,9,10,11]. These adaptative processes, known as phosphate starvation responses (PSRs), are categorized into local and systemic Pi responses as Pi sensing and long-distance signaling, and the adaptative efficiency can be determined by the PSR regulatory capacity in plants [6,10,12,13].

The local Pi responses are largely associated with traits for root growth, as they perceive the Pi level in the soil and determine the root developmental fate in response to the imposed Pi-starvation stress, whereas internal Pi concentration activates the systemic responses to integrate the Pi level information in distantly located plant tissues/organs [6,10,12,13]. Systemic Pi responses play an important role in global P homeostasis within plants, and microRNA399 (miR399) was the first identified long-distance regulatory component in Pi-stress signaling [14,15]. Under Pi-starvation conditions, miR399 is expressed in shoots and delivered into roots via the phloem to mediate the degradation of *PHO2* mRNA, which encodes an E2 ubiquitin-conjugating enzyme and regulates the level of high-affinity PHOSTPHATE TRANSPORTER1 (PHT1) protein in roots. Therefore, this systemic regulatory mechanism serves to control Pi acquisition under Pi-starvation stress conditions [6,14,15,16,17,18,19,20].

MYB transcription factors (TFs), one of the largest TF families in plants, contain a conserved DNA-binding domain and up to three tandem repeats of helix-turn-helix domains, designated R1, R2 and R3 [21,22,23]. In plants, MYB44, classified as a member of the R2R3-type MYB subfamily, has been characterized as a regulatory factor in various abiotic and biotic stress responses [24,25,26,27,28]. It was earlier reported that MYB TFs are involved in the control of Pi-starvation responses in plants [28,29,30,31,32,33,34]. PHOSPHATE STARVATION RESPONSE1 (PHR1) is a MYB-related TF, a key regulator in Pi-starvation responses [29]. Arabidopsis MYB62, induced in response to Pi-stress, appears to function as a negative regulator for PSR gene expression and is involved in gibberellic acid biosynthesis [31]. Another MYB-like TF, REGULATOR OF LEAF INCLINATION 1 (RLI1), interacts with SPX1 (Syg1/Pho81/XPR1), and the SPX1-RLI1 complex appears to block RLI1 binding to the promoter regions of RLI1 target genes, which are involved in elongation of leaf cells in the lamina joint in rice [33,34]. Additionally, potato MYB44 functions as a negative regulator of *PHOSPHATE1* (*PHO1*) expression, which encodes a major Pi transport protein that mediates Pi transport from the root to the shoot, through Pi loading into the xylem [28]. However, it remains unknown whether *MYB44* plays a role in systemic Pi responses. In this study, Arabidopsis *MYB44*, referred to as *AtMYB44*, was characterized as a phloem-mobile mRNA, and shown to be involved in systemic regulation of *PHT1* (Pi transporter) gene expression in roots under Pi-starvation conditions. We propose that mobile *AtMYB44* mRNA acts as a negative regulator in systemic Pi responses for control of Pi acquisition in Arabidopsis roots.

## 2. Results

### 2.1. Identification of Mobile CsMYB44 Orthologs in Arabidopsis

In a previous study, cucumber was used as a model plant to examine the role of the phloem in systemic signaling under Pi-stress conditions [35]. The phloem mRNA profiles changed dynamically, in response to an imposed Pi-stress treatment, and a range of graft-transmissible mRNAs were shown to target specific sink organs/tissues under Pi-stress conditions [35]. The *Cucumis sativus MYB44* (*CsMYB44*, Csa6G491690), encoding an R2R3 MYB transcription factor (TF), was among these Pi-stress-induced graft-transmissible mRNAs and was delivered into sink tissue during the early stage of Pi-stress treatment [35]. Interestingly, even though 112 MYB family members were identified in cucumber, [36] and one specific clade included four other cucumber MYBs with CsMYB44 (Appendix A), *CsMYB44* (*Csa6G491690*) was detected as the only Pi-stress-responsive mobile mRNA, encoding for a MYB transcription factor [35]. Based on our findings, we hypothesize that mobile *CsMYB44* mRNA could serve as an early, long-distance signaling factor for systemic Pi responses. Thus, we selected *CsMYB44* to further assess its role as a long-distance signaling mRNA in response to Pi-stress.

First, to facilitate the testing of our hypothesis that *MYB44* plays a role in phloem-mediated systemic signaling under Pi-starvation stress conditions, we conducted a phylogenetic analysis to identify putative *CsMYB44* orthologs in Arabidopsis. A total of 132 Arabidopsis R2R3 MYB family members were analyzed (Appendix A), and here, the phylogenetic tree analysis revealed the presence of CsMYB44 in a specific MYB subfamily subclade (Figure 1A and Appendix A). AtMYB77, AtMYB73 and AtMYB70 shared an amino acid sequence identity of 69.0%, 59.5% and 57.8%, respectively, with AtMYB44.

To gain insight into the regulation of the genes in this MYB subclade, we used the Arabidopsis cis-regulatory element database (https://agris-knowledgebase.org/AtcisDB; accessed on 4 February 2020) to analyze the promoter regions of *AtMYB44*, *AtMYB70*, *AtMYB73* and *AtMYB77* with the aim to identify potential *cis*-elements that might serve to control their expression under Pi-stress conditions. Numerous Pi-starvation responsive (PSR) genes are regulated by PHR1, and the PHR1 binding site (P1BS) is enriched in the promoter regions of many Arabidopsis Pi-responsive genes [37,38,39].

Our *cis*-element analysis indeed revealed that P1BS motifs were located within the promoters of *AtMYB44*, *AtMYB70* and *AtMYB73*, but not *AtMYB77* (Figure 1B). Furthermore, qRT-PCR analysis established that the transcript levels of *AtMYB44*, *AtMYB70*, *AtMYB73* and *AtMYB77* were responsive to a Pi- stress treatment in both shoots and roots (Figure 1C,D). For *AtMYB44* and *AtMYB77*, at least a two-fold increase and decrease in transcript levels, respectively, were detected in shoots, under Pi-stress conditions (Figure 1C). By contrast, *AtMYB70* and *AtMYB73* transcript levels in the shoots were only slightly changed in response to Pi-stress (10 µM Pi) (Figure 1C). In roots, the *AtMYB44*, *AtMYB70*, *AtMYB73* and *AtMYB77* transcript levels were significantly increased under Pi-stress conditions (10 µM Pi) (Figure 1D). Taken together, our results demonstrate that the putative CsMYB44 homologs, *AtMYB44*, *AtMYB70*, *AtMYB73* and *AtMYB77*, are regulated in response to the imposed Pi-stress treatment. Interestingly, the previous report provided evidence that *AtMYB44* and *AtMYB70* mRNA were mobile and, in particular, Pi-stress appeared to lead to phloem mobility of *AtMYB44* mRNA in Arabidopsis [40]. In addition, *AtMYB44* transcript levels increased in both shoot and root (Figure 1C,D); therefore, we focused on investigating the role of *AtMYB44* in Pi-stress systemic signaling.

### 2.2. AtMYB44 Is Expressed in Leaf and Root Vascular Tissues under Pi-Stress Conditions

To further understand the mechanism of *AtMYB44* regulation in Arabidopsis, we generated transgenic plants carrying the GFP-GUS reporter under the control of the native *AtMYB44* promoter and analyzed the *AtMYB44* expression pattern (Figure 2). Consistent with our qRT-PCR results (Figure 1C,D), the β-glucuronidase (GUS) staining assays showed signal in both shoots and roots, and revealed stronger staining under Pi-stress (10 µM) compared with Pi-sufficient conditions (500 µM) (Figure 2A,B), consistent with induction of *AtMYB44* expression in response to Pi-stress. Here, *AtMYB44* expression was detected in leaf lamina, leaf vasculature, root vasculature, root tips of primary and lateral roots, and in lateral root primordia (Figure 2B–J). Some mature lateral roots also exhibited GUS staining in the vascular tissue of elongation and maturation zones, but not the root tip (Figure 2H, compared with Figure 2G), suggesting that the expression of *AtMYB44* would respond to Pi-starvation stress at the lateral root tips. Within the primary root tip, *AtMYB44* was highly expressed in the epidermis, cortical layers and endodermis under Pi-starvation conditions (Figure 2K,L). In time-course experiments, the effects of Pi-starvation stress were confirmed on *AtMYB44* expression in shoots and roots in Arabidopsis (Figure 2M,N and Appendix A). These findings were consistent with the notion that *AtMYB44* plays a role in transcriptional regulation of adaptive shoot and root development in response to Pi-stress treatment.

### 2.3. AtMYB70, AtMYB73 and AtMYB77 Transcript Levels Were Elevated in atmyb44 Mutant, Compared with Wild-Type

As expression of other *AtMYB* family genes, close to CsMYB44, was also responsive to imposed Pi-stress (Figure 1D), we hypothesized that other redundant AtMYBs could compensate for the absence of *AtMYB44* under Pi-stress conditions. To test this notion, we examined the expression levels of *AtMYB70*, *AtMYB73* and *AtMYB77* in the *atmyb44* mutant background under Pi-stress conditions using qRT-PCR (Figure 3). Here, we confirmed that the expression of *AtMYB70*, *AtMYB73* and *AtMYB77* increased (Figure 3). Enhanced transcript levels of *AtMYB70*, *AtMYB73* and *AtMYB77* were detected in *atmyb44* plants under both Pi-sufficient and Pi-stress conditions, compared with WT (Figure 3). These assays supported a role for *AtMYB70*, *AtMYB73* and *AtMYB77* in the functional compensation of *AtMYB44* in its absence.

### 2.4. The atmyb44 Root Had an Elevated Pi Concentration under Pi-Stress Conditions

Under Pi-sufficient conditions, the *atmyb44* mutant exhibited slightly longer primary roots and a higher number of lateral roots, compared with the WT and *AtMYB44* overexpression (OX) lines, and, consistent with the previous study [25,41], *AtMYB44* overexpression (OX) lines displayed shorter primary roots than WT (Appendix A). However, under Pi-stress conditions, no significant differences in primary root growth were observed between WT, *atmyb44* and *AtMYB44* OX lines (Appendix A). Interestingly, the lateral root numbers of *AtMYB44* OX lines were lower than in WT and *atmyb44* mutant lines, and this was observed in both Pi-sufficient and Pi-starvation conditions (Appendix A). Under Pi-stress conditions, the *atmyb44* plants did not exhibit a clear phenotypic difference in primary root growth and lateral root number, compared with WT (Appendix A). As both plant lines were grown on agar medium, which limited observation of plant developmental changes, to apply a prolonged Pi-stress, we next employed a hydroponic system to conduct phenotypic analysis of *atmyb44* and *AtMYB44* overexpression (OX) lines. In the experiments using the hydroponic system, we reduced the Pi concentration from 10 µM to 5 µM to ensure we were creating effective Pi-starvation treatments in hydroponics, where Pi is more available than in agar growth medium.

Compared to WT, around a 1.2-fold higher and a 1.5-fold lower shoot biomass was observed in the *atmyb44* and *AtMYB44* OX plants, respectively, under Pi-sufficient conditions, although no significant difference in shoot biomass was detected under Pi-stress conditions (Figure 4A and Appendix A). However, the root biomass was obviously at least 1.5-fold higher in *atmyb44* than WT and *AtMYB44* OXs, under both Pi-sufficient and Pi-stress conditions (Figure 4B and Appendix A). In *atmyb44*, the Pi concentration was around 2-fold higher in shoots and 1.5-fold lower in roots compared to the level in WT under Pi-sufficient conditions. In contrast, the root Pi concentration in *atmyb44* was around 2-fold higher than WT under Pi-stress conditions, even though no significant difference in shoot Pi concentration was detected between WT and *atmyb44* (Figure 4C,D). Taken together, these results suggested that AtMYB44 functions as a negative regulator in both shoot and root development under Pi-sufficient conditions and that an absence of *AtMYB44* might increase Pi uptake into Arabidopsis roots.

### 2.5. Expression of PHT1;2 and PHT1;4 Is Negatively Regulated by AtMYB44 in Roots

The in silico analysis using ConnecTF (https://connectf.org; accessed on 23 October 2022) [42] revealed that AtMYB44 could be involved in many gene regulatory pathways and, interestingly, bound to the promoter regions of *PHOSPAHTE TRANSPORTER 1;2* (*PHT1;2*) and *PHT1;4* as potential AtMYB44-target sites (Appendix A). As we observed an enhanced level of Pi concentration in the root of *atmyb44* plants under Pi-stress conditions (Figure 4D), and PHT1 regulates the initial uptake of Pi [6,43,44,45,46,47,48,49,50], we hypothesized that AtMYB44 regulates *PHT1* expression in roots for Pi acquisition under Pi-stress conditions.

To test this hypothesis, we examined the expression level of *PHT1;2* and *PHT1;4* in WT, *atmyb44* and *AtMYB44* OX-1 roots, using qRT-PCR (Figure 5). The expression levels of *PHT1;2* and *PHT1;4* were much higher in *atmyb44* under Pi-stress conditions compared to WT (two- to four-fold higher expression; Figure 5), and the increased levels of *PHT1;2* and *PHT1;4* in *atmyb44* were diminished in *AtMYB44* OX-1 (Figure 5A,B). We did not detect any significant differences in the expression levels of *PHT1;2* in *AtMYB44* OX-1, compared with WT (Figure 5A). The Type B *MONOGALACTOSYL DIACYLGLYCEROL SYNTHASE 3* (*MGD3*) has been reported as a Pi-starvation response gene [51]; however, it was not predicted to be a AtMYB44 target (Appendix A). Compared with the expression patterns of *PHT1;2* and *PHT1;4*, no significant difference in *MGD3* expression was observed between WT, *atmyb44* and *AtMYB44* OX-1 (Figure 5C). It is plausible that AtMYB44 might target the promoter of *PHT1;2* and *PHT1;4*, where it acts as a negative regulator.

### 2.6. Mobile AtMYB44 mRNA Functions as a Negative Regulator of PHT1;2 and PHT1;4 Expression

We demonstrated that *AtMYB44* expression responds to Pi-stress (Figure 1C,D) and AtMYB44 appears to play a role in regulating *PHT1;2* and *PHT1;4* expression under these conditions (Figure 5). As *AtMYB44* was identified as a mobile *CsMYB44* ortholog in Arabidopsis (Figure 1A), it is plausible that *AtMYB44* could serve as a long-distance signal to exert control over Pi-starvation responses in Arabidopsis. To test whether *AtMYB44* acts as a systemic signal in response to Pi-stress, we employed a split-root system to test whether *AtMYB44*-mediated gene expression is a systemic or local response under Pi-stress conditions. The Arabidopsis root systems grown in the hydroponic culture system were separated into two parts with each half of the root system placed into a vessel containing nutrient solution with Pi (200 µM) or with no Pi (0 µM). As the control, both vessels for each half of the root system on a plant contained the same nutrient solution (CP200 for 200 µM Pi, CP0 for 0 µM) to mimic plant growth under homogenous Pi-sufficient or deficient conditions (Figure 6A) [12].

Total RNA was extracted from these roots for qRT-PCR analysis. First, we investigated regulation of *AtMYB44* expression to see if it is part of a systemic response, after Pi-sufficient and deficient treatments were applied to different halves of the root system on individual plants area (Figure 6). As expected, no *AtMYB44* transcript was detected in *atmyb44* tissues, and *AtMYB44* expression in WT was enhanced in response to the imposed Pi-stress (CP0), compared with the mRNA level under Pi-sufficient conditions (CP200) (Figure 6B). In WT, a higher level of *AtMYB44* mRNA was detected in that half of a plant’s root system placed in the compartment with 200 µM Pi (SP200), relative to the control (CP200). In addition, lower *AtMYB44* mRNA levels were observed for the split root system in 0 µM Pi (SP0) compared to the transcript level in the homogenously Pi-deficient control (CP0) (Figure 6B). These findings support our hypothesis that *AtMYB44* serves as a systemic regulator.

Next, we examined systemic regulation of *PHT1;2* and *PHT1;4* expression in *AtMYB44* knockout mutants again using this split-root system to test whether *AtMYB44* serves as a factor in systemic *PHT1;2* and *PHT1;4* regulation. Here, the qRT-PCR analysis revealed that, consistent with the previous report [12], expression of both *PHT1;2* and *PHT1;4* in WT plants was downregulated in roots within the SP0 container, compared to *PHT1;2* and *PHT1;4* expression in CP0 in WT (Figure 6C,D). However, in *atmyb44* plants, the enhanced transcript levels of *PHT1;2* and *PHT1;4* were decreased in SP0 compared to CP0 (Figure 6C,D). Taken together, these results suggest that *AtMYB44* expression is regulated in a systemic manner, and that other long-distance regulators might be involved in the systemic regulation of *PHT1;2* and *PHT1;4* regulation in the *atmyb44* background.

To investigate whether *AtMYB44* mRNA acts as a long-distance regulator for control over *PHT1;2* and *PHT1;4* expression, we performed micrografting assays between *atmyb44* and *AtMYB44* OX-1 lines (Figure 7A). Here, our RT-PCR analyses revealed that, as controls, *AtMYB44* expression was increased in *AtMYB44* OX-1, but not in *atmyb44*, and *AtMYB44* OX-1 and *atmyb44* were subsequently used as shoot scions or root stocks (Figure 7B). Analysis of these grafted tissues revealed that the transgene *AtMYB44* was detected in rootstocks of heterografted *AtMYB44* OX-1 (scion)/*atmyb44* (rootstock), but not in the *atmyb44* (scion)/ *AtMYB44* OX-1 (rootstock), consistent with *AtMYB44* mRNA being phloem-mobile from source to sink tissues (Figure 7B). Interestingly, the level of *PHT1;2* and *PHT1;4* expression was decreased in the rootstocks of these heterografted *AtMYB44* OX-1 (scion)/*atmyb44* (rootstock) plants, compared to the *atmyb44* homograft (Figure 7C). These data support the hypothesis that *AtMYB44* mRNA acts as a mobile, negatively acting regulator of *PHT1;2* and *PHT1;4* expression in Arabidopsis roots.

## 3. Discussion

The local and systemic regulatory mechanisms acting in Pi acquisition and distribution under Pi-stress conditions have been extensively studied, and many important genes have been identified as crucial components that regulate Pi homeostasis in plants, such as Arabidopsis and rice [4,5,6,29,48,50,52,53]. However, limited information is available on the nature of the systemic Pi-signaling agents that control Pi homeostasis at the whole-plant level [6,14,15,17,35,54,55,56,57]. In this study, we identified *AtMYB44* mRNA as a potential mobile systemic Pi-signaling component in Arabidopsis.

### 3.1. AtMYB44 Expression Responds to an Imposed Pi-Stress

Four homologs of CsMYB44 were identified as potential CsMYB44 orthologs in Arabidopsis (Figure 1A and Appendix A), and their expression patterns were also responsive to an imposed Pi-stress (Figure 1B–D). As the promoter regions of *AtMYB44*, *AtMYB70* and *AtMYB73*, but not *AtMYB77*, included PIBS motifs, this suggested that *AtMYB44*, *AtMYB70* and *AtMYB73* expression could be regulated in a PHR1-dependent manner under Pi-stress conditions.

An increase in the level of *AtMYB44* expression during Pi-starvation treatment and detection of GUS signals within the vascular tissues of both shoot and root (Figure 2) suggested that AtMYB44 could function within the vasculature in response to Pi-starvation stress. In addition, we also observed that *AtMYB44* expression was strongly detected in the root tips and lateral root primordia (Figure 2D,G–L). Based on our in silico analysis, the AtMYB44 TF could be associated with the promoter regions of many *AUXIN RESPONSE FACTORs* (*ARFs*), which contribute to control over *PHR1* expression in Arabidopsis roots under Pi-starvation conditions [58,59] (Appendix A). As *AtMYB44* expression is responsive to auxin levels in the root [60], AtMYB44 may participate in auxin-dependent root development in response to Pi-stress.

AtMYB44, AtMYB70, AtMYB73 and AtMYB77 share high structural similarity and participate in root system development [24,61,62]. Interestingly, we detected enhanced expression of *AtMYB70*, *AtMYB73*, and *AtMYB77* in *atmyb44* plants, compared with WT (Figure 3). It was earlier proposed that AtMYB44 is a functional paralog of AtMYB73 and AtMYB77 for auxin-mediated lateral root growth and development [61]. Even though the functional redundancy of AtMYB70 with AtMYB44 in root growth has not yet been examined, it is implicit that AtMYB70, AtMYB73 and AtMYB77 are redundant with AtMYB44 to coordinate adaptive root development under Pi-stress conditions.

As a previous report shows [25], the *atmyb44* root appears to have slightly enhanced root growth performance under both Pi-sufficient and Pi-stress conditions compared to WT (Figure 4A,B, Appendix A). One proposed AtMYB44 function is as a negative regulator in abscisic acid (ABA) responses, which are involved in root growth inhibition [25,63,64]. AtMYB44 interacts with PYRABACTIN RESISTANCE 1-LIKE 8 (PYL8), identified as an ABA receptor, to regulate early ABA signaling and promote lateral root growth [25,61]. ABA signaling can play a role in enhancing the promoter activity of various auxin-responsive genes [64]. Although limited information is available regarding the role of ABA in Pi-starvation responses [65], it is plausible that AtMYB44 might inhibit ABA responses and activate auxin signaling to enhance root development in response to the imposed Pi-stress. However, our *AtMYB44* OX lines did not show a clearly opposite phenotype with the *atmyb44* line under Pi-stress, even though, based on the previous report, the expression level of MYB44 was 23- and 12-fold higher in *AtMYB44* OX-1 and 2 lines compared with WT [25,33] (Figure 4, Appendix A). This result could not be explained as a role of AtMYB44 in Pi-starvation signaling; however, as AtMYB44 has been proposed to be involved in multiple signaling pathways, ectopic expression of *AtMYB44* might result in aberrant ABA and auxin signaling to establish unidentified molecular responses under these Pi-stress conditions.

### 3.2. AtMYB44 mRNA Acts as a Systemic Pi Signaling Factor to Negatively Regulate Root Pi Transport Systems

In this study, we provide insight into the function of AtMYB44 as a negative regulator in systemic responses to Pi-stress, which might be associated with the promoter regions of *PHT1;2* and *PHT1;4* (Appendix A). Enhanced *PHT1;2* and *PHT1;4* expression and elevated soluble Pi concentration were detected in *atmyb44* plants, compared to WT (Figure 4C,D and Figure 5). It is noticeable that the soluble Pi concentration was higher in shoots and lower in roots of the *atmyb44* mutant under Pi-sufficient conditions, compared with WT (Figure 4C,D). This suggests that, in Arabidopsis, AtMYB44 plays a negative role in root-to-shoot Pi transport under Pi-sufficient conditions. Hence, Pi translocation through the xylem might be enhanced in *atmyb44* plants. Future research is needed to provide relevant evidence to verify the detailed binding mechanisms between AtMYB44 and the promoter regions of *PHT1;2* and *PHT1;4* under both Pi-sufficient and Pi-starvation conditions.

Interestingly, under Pi-stress conditions, although a similar level of soluble Pi concentration was detected in the shoots of WT and *atmyb44*, its level was higher in *atmyb44* roots, and this result seems to be correlated with increased root biomass (Figure 4 and Appendix A). Additionally, shoot biomass of *atmyb44* was significantly elevated under Pi-sufficient, but not Pi-limiting conditions (Figure 4). It is possible that AtMYB44 functions in roots as a negative regulator of *PHT1;2* and *PHT1;4* (Figure 5); thus, enhanced levels of PHT1;2 and PHT1;4 would increase the efficiency of Pi uptake and plant growth performance in *atmyb44* plants under Pi-stress conditions. Taken together, it appears that AtMYB44 might play a negative role in Pi transport from root to shoot under Pi-sufficient conditions, but during a Pi-stress treatment, it might act as a negative regulator on Pi uptake in Arabidopsis.

The split-root assays demonstrated that *AtMYB44* expression is systemically regulated by Pi-stress (Figure 6B). Consistent with a previous study [12], systemic responses of *PHT1;2* and *PHT1;4* were observed, and interestingly, such long-distance regulation was partially disrupted in the *atmyb44* mutant background (Figure 6C,D). The mobility of *AtMYB44* mRNA appeared to reduce the level of *PHT1;2* and *PHT1;4* transcript abundance in roots (Figure 7). In this regard, various RNA species, including mRNAs, small interfering RNAs, and non-coding RNAs, etc., have been shown to be translocated through the phloem, and some mobile RNA molecules can act as signaling agents in plant development and physiology [14,35,40,55,56,66,67,68,69,70,71]. For example, *GIBBERELLIC ACID-INSENSITIVE* mRNA is transported from the source to sink tissues and contributes to regulating leaf development [72,73]. Additionally, potato tuberization is regulated with the transportation of *SP6A* mRNA from potato leaves to the underground stolon [74,75]. The shoot-derived *INDOLEACETIC ACID18* (*IAA18*), *IAA28* and *TRANSLATIONALLY CONTROLLED TUMOUR PROTEIN* (*TCTP*) traffic into the roots to regulate the lateral root development [76,77,78].

In our study, we propose that mobile *AtMYB44* mRNA serves as a long-distance phloem-based signal in the Pi-stress response (Figure 7). Although *AtMYB44* lacked modified base 5-methylcytosine (m5C), CU- or tRNA-like motifs, which have been characterized as contributing to phloem mRNA mobility [79,80,81], our heterografting assay established the mobility of *AtMYB44* mRNA in Arabidopsis (Figure 7 and Appendix A). As AtMYB44 and CsMYB44 protein have not been detected in the phloem exudate [82,83], it is likely that *AtMYB44* mRNA and not protein serves as the signaling agent in Pi homeostasis. Although mobile *AtMYB44* mRNA is involved in regulating *PHT1* expression, in roots, the regulatory mechanism underlying its phloem mobility, in Pi acquisition and utilization, remains to be elucidated. Based on previous studies [80,81], the shoot-derived *AtMYB44* mRNA could be transported through the phloem and then translated in the targeted root tissues to regulate Pi transport.

Our study showed enhanced *AtMYB44* expression under Pi-stress conditions, and a negative role for AtMYB44 in *PHT1* expression in Arabidopsis roots (Figure 8). As an increase in both *AtMYB44* and *PHT1* expression was detected under Pi-stress, this raises the question as to why plants would enhance *AtMYB44* expression to repress the Pi transport system under Pi-stress conditions, even though one would expect these plants to increase their Pi uptake capacity to adapt under limited Pi input. Mobile *AtMYB44* mRNA might play a role in fine-tuning the regulation of Pi homeostasis in response to Pi-stress for control over the energy balance between adaptive plant development and Pi uptake (Figure 8). Our current findings provide insight into plant Pi-stress regulatory pathways and further studies will be required to reveal the molecular mechanism by which such mobile *AtMYB44* mRNA effect the ability of the plant to acquire and utilize Pi during Pi-stress conditions. Such information would be of value in genetic engineering of crops for improved yield performance under reduced Pi fertilizer applications.

## 4. Materials and Methods

### 4.1. Plant Materials

Arabidopsis plants were grown in controlled environment chambers under long-day conditions (16 h light/8 h dark, 120–150 μmol m^−2^ s^−1^ of photosynthetically active radiation), at temperatures of 22 °C day/18 °C night [84,85]. The T-DNA knock-out line (SALK_039074 [*atmyb44*]) for AT5G67300 was obtained from the Arabidopsis Biological Resource Center (ABRC, Columbus, OH, USA). The genotypes were confirmed using PCR analysis with appropriate primer sets (Appendix A).

### 4.2. Growth Conditions and Pi-Stress Treatments

Arabidopsis plant lines were grown on solid medium (pH 5.7), in sand, or in a hydroponic culture system, as described previously [35,84,86,87,88,89]. In brief, Arabidopsis seeds were sterilized, germinated on solid medium with 500 µM NH_4_H_2_PO_4_ and then, at 7 days after germination (DAG), they were transferred onto fresh medium or medium with (NH_4_)_2_SO_4_ that partially or completely replaced NH_4_H_2_PO_4_. For hydroponically grown Arabidopsis plants, the tip of a 10 µL pipette was cut and then filled with 0.7% (*w*/*v*) agar, upon which sterilized seeds were sown, prior to setting them up on a floating board system. Hydroponic solution was replaced every 5 days. The solution for Pi-sufficient and starvation treatment contained 200 µM and 5 µM of NH_4_H_2_PO_4_, respectively. For growing Arabidopsis plants in sand, seeds were germinated on solid medium with 500 µM (Pi-sufficient) or 10 µM (Pi-starvation) of NH_4_H_2_PO_4_ for 14 days, followed by transplanting to silica sand. Arabidopsis plants were further grown in silica sand for 14 days with nutrient solution, which contained 500 µM (Pi-sufficient) or 10 µM (Pi-starvation) of NH_4_H_2_PO_4_, supplied to the roots. Images of root system architecture were captured with a Nikon D7200 (Nikon, Tokyo, Japan) digital camera. Acquired images were processed and analyzed using ImageJ software (version 1.53), as described previously [89].

Shoot and root tissues were collected from hydroponically cultured Arabidopsis plants and were used to measure biomass and Pi concentration, as described previously [86,87]. Collected shoot and root tissues were frozen in liquid nitrogen after measurement of fresh weight and then homogenized using a bead beater. Ground samples were digested with 5 M H_2_SO_4_, and then Pi concentrations were determined using a continuous flow analyzer (Skalar, Breda, The Netherlands), following the manufacturer’s instructions.

Four-week-old Arabidopsis plants were used for the split-root assays, as described previously [12,90], with modifications. Arabidopsis roots were divided into two parts and then immersed in two separate compartments, which contained either the same nutrient solution with 200 µM Pi (CP200) or 0 µM Pi (CP0), or a different nutrient solution with 200 µM Pi (SP200) and 0 µM Pi (SP0), for 7 days [12,90].

### 4.3. Phylogenetic Analysis

The amino acid sequences of AtMYBs were obtained from the Arabidopsis Information Resource Database (https://www.arabidopsis.org/, 4 November 2019) and aligned using MEGA 11 software. The phylogenetic tree was constructed using a neighbor-joining (NJ) method with 1000 bootstrap replicates.

### 4.4. β-Glucuronidase (GUS) Histochemical Analysis and Confocal Microscopy

The 2916 bp upstream region, which was from the ATG start codon of *AtMYB44*, was amplified from genomic DNA of Arabidopsis WT (Col-0), using the primer set listed in Appendix A. The fragment was then inserted into the TOPO-D vector (Thermo Fisher Scientific, Waltham, MA, USA), followed by an LR clonase reaction (Invitrogen), to subclone the *AtMYB44* promoter into pBGWFS7. The construct was introduced into *Agrobacterium tumefaciens* strain GV3101 to generate transgenic Arabidopsis plants. The GUS staining of transgenic plants, carrying a *GUS* reporter gene under the control of the *AtMYB44* native promoter, was conducted, as described previously [84]. In brief, 12-day-old transgenic plants were immersed in 50 mM of sodium phosphate buffer (pH 7.0) with 1 mM K_3_Fe(CN)_6_, 1 mM K_4_Fe(CN)_6_, and 2 mM 5-bromo-4-chloro-3-indolyl β-D-glucuronide sodium salt and 10 mM EDTA (pH 8.0) and incubated at 37 °C. The chlorophyll was eliminated using 70% ethanol. T3 plants collected from three independent transgenic lines were used for GUS assays and imaged, using a THUNDER stereomicroscope (Leica, Wetzlar, Germany) and Zeiss Axioskop 2 Plus microscope.

For the confocal microscopy, propidium iodide (PI) (10 mg/mL) was used to stain the roots of an Arabidopsis seedling. GFP and PI were excited with 488 nm light, and emission was detected at 505–545 nm for GFP and 605–635 nm for PI. At least 6 Arabidopsis seedlings were examined for the confocal microscopy, and the same scanning conditions were used to take all images using Leica Stellaris 5.

### 4.5. Micrografting

Arabidopsis micrografting was performed, as described previously [76,91]. Briefly, hypocotyls of 5-day-old Arabidopsis seedlings, grown on 1/2 × MS medium with 0.8% agar, were cut on nylon membranes using a surgical blade. The scion and stock were placed together to connect the shoot and root. Grafted Arabidopsis plants were grown on 1/2 × MS medium with 1.5% agar for 5 days. Successfully grafted plants were transferred to 1/2 × MS medium with 0.8% agar for further studies.

### 4.6. RNA Extraction and qRT-PCR

Total RNA was extracted from Arabidopsis leaves and roots, using the TRIzol^®^ Reagent (Thermo Fisher Scientific, Waltham, MA, USA), following the manufacturer’s instructions. Total RNA of 1 µg was used for cDNA synthesis with the SuperScript IV first-strand synthesis system (Invitrogen). The qRT-PCR analysis was conducted, as described previously [92]. Briefly, the qRT-PCR was performed with PowerUp™ SYBR™ Green Master Mix (Thermo Fisher Scientific, Waltham, MA, USA), using the QuantStudio™ 6 Flex Real-Time PCR Systems (Life Technologies), to detect *AtMYB44*, *AtMYB70*, *AtMYB73*, *AtMYB77*, *PHT1;2*, *PHT1;4*, and *MGD3* with the primer sets listed in Appendix A. *AtActin* was used as a reference gene for normalization of transcript levels. Transcript levels and ratios were calculated using the 2^−∆Ct^ or the 2^−∆∆Ct^ method, respectively. Statistical analyses were performed with the Student’s t-test and Tukey’s HSD (honestly significant difference) test. At least three biological and three technical replicates were used for qRT-PCR analyses.

## Figures and Tables

**Figure 1 plants-12-03617-f001:**
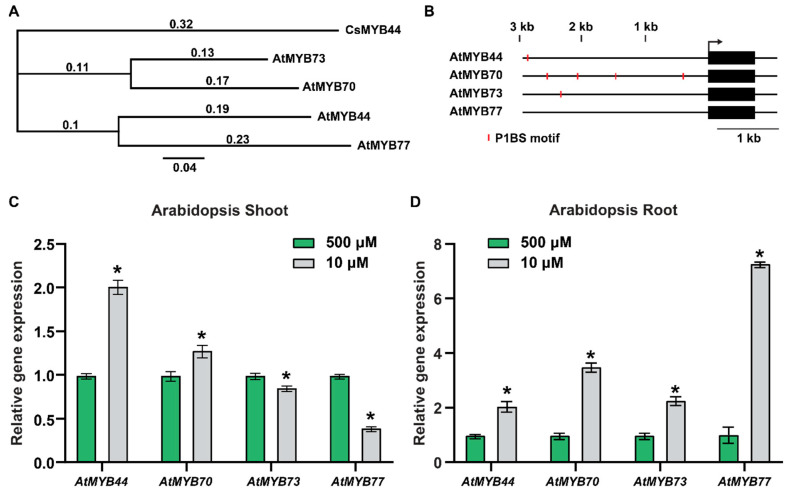
AtMYB44 is a potential functional homolog of CsMYB44. (**A**) The clade, which includes the closest MYB homologs of CsMYB44, in the phylogenic tree of the Arabidopsis MYB family, is shown in Appendix A. Numbers above the phylogenetic tree indicate posterior probabilities. (**B**) P1BSs, which are PHR1 binding motifs, predicted in the promoter regions of *AtMYB44*, *AtMYB70* and *AtMYB73*, but not *AtMYB77*. Red bars indicate the P1BS motifs. (**C**,**D**) Relative expression of *AtMYB44*, *AtMYB70*, *AtMYB73*, and *AtMYB77* in the shoot (**C**) and the root (**D**), under Pi-sufficient (500 µM) and Pi-starvation (10 µM) conditions. Arabidopsis seedlings were transplanted onto the medium with 500 µM, or 10 µM Pi, 5 days after germination on solid medium with 500 µM Pi. Shoot and root samples of Arabidopsis plants were harvested 7 days after Pi-sufficient (500 µM), or Pi-starvation (10 µM) treatment. Arabidopsis *Actin* was used as an internal control to normalize the qRT-PCR results. The data are presented as mean ± SD (three technical replicates and three technical repeats). Asterisks indicate significantly different values between Pi-sufficient and Pi-starvation conditions (Student’s *t*-test, *p* < 0.05).

**Figure 2 plants-12-03617-f002:**
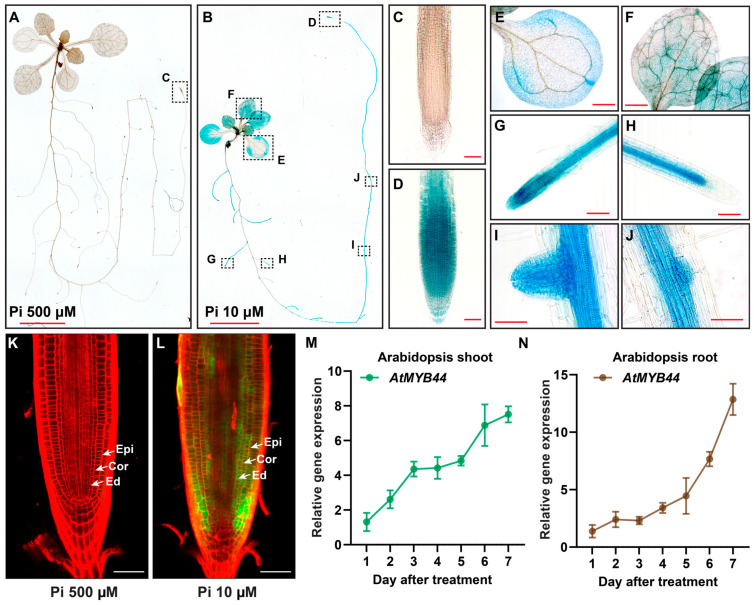
*AtMYB44* is expressed in response to Pi-starvation stress. Histochemical staining in 12-day-old transgenic Arabidopsis plants expressing the GFP-GUS reporter under the control of the *AtMYB44* native promoter. Transgenic Arabidopsis seeds were germinated on solid medium with 500 µM Pi (Pi-sufficient medium) for 5 days and then transplanted to medium with 500 µM, or 10 µM Pi. Arabidopsis seedlings were collected 7 days after Pi-sufficient (500 µM), or Pi-starvation (10 µM) treatment. (**A**) GUS expression was barely detectable in Arabidopsis seedlings grown under Pi-sufficient (500 µM) conditions. (**B**) Under Pi-starvation (10 µM) conditions, GUS expression was observed in the primary root tip ((**D**), compared with (**C**) which showed GUS expression in the primary root tip under Pi-sufficient [500 µM] conditions), and Pi starvation also enhanced GUS expression in cotyledons (**E**), developing true leaves (**F**), lateral roots (**G**,**H**), and developing lateral root primordia (**I**,**J**). Dotted boxes indicate magnified regions for (**C**–**J**). Bars: 5 mm in (**A**,**B**), 500 µm in (**E**,**F**), and 50 µm in (**C**,**D**,**G**–**J**). In (**K**,**L**), *AtMYB44* expression was increased in the epidermis, cortex layer and endodermis of the primary root tip under Pi-starvation (10 µM) conditions. In these representative confocal images, the GFP signal was enhanced in the primary root tip in response to the Pi starvation treatment (10 µM) (**L**) compared to growth on Pi-sufficient (500 µM) media (**K**) for 7 days. Arabidopsis seedling roots were stained with propidium iodide (PI, red). Epi, epidermis; cor, cortex layer; end, endodermis. Bar = 50 µm. (**M**,**N**): *AtMYB44* expression was increased in both shoots (**M**) and roots (**N**) under Pi-starvation conditions. Arabidopsis seedlings (Col-0), germinated on agar medium with 500 µM Pi (Pi-sufficient medium) for 5 days, were transplanted to medium with 500 µM, or 10 µM Pi, and then collected over a 7-day time course of Pi-starvation treatment. The Arabidopsis *Actin* gene was used as an internal control to normalize the qRT-PCR results. The data are presented as mean ± SD (three biological replicates and three technical repeats).

**Figure 3 plants-12-03617-f003:**
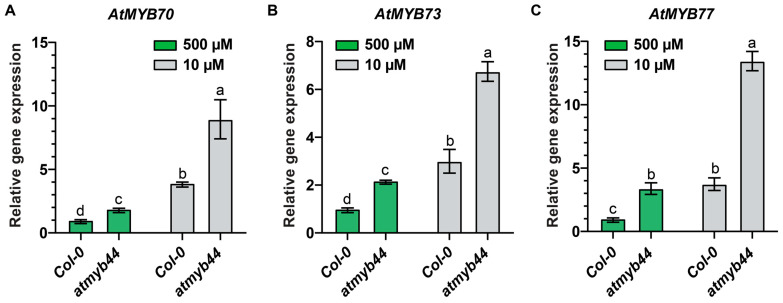
Transcript levels of *AtMYB70*, *AtMYB73*, and *AtMYB77* are enhanced in *atmyb44* knock-out mutants. Relative expression levels of *AtMYB70* (**A**), *AtMYB73* (**B**), and *AtMYB77* (**C**) in WT (Col-0) and the *atmyb44* mutant under Pi-sufficient (500 µM) and Pi-starvation (10 µM) conditions. Arabidopsis seedlings were transplanted onto medium with 500 µM, or 10 µM Pi 5 days after germination on solid medium with 500 µM Pi. Arabidopsis seedlings were harvested 7 days after Pi-sufficient (500 µM) or Pi-starvation (10 µM) treatment. Arabidopsis *Actin* was used as an internal control to normalize the qRT-PCR results. The data are presented as mean ± SD (three technical replicates and three technical repeats). The different lowercase letters indicate significant differences in relative gene expression levels, determined by the Tukey’s test (*p* < 0.05).

**Figure 4 plants-12-03617-f004:**
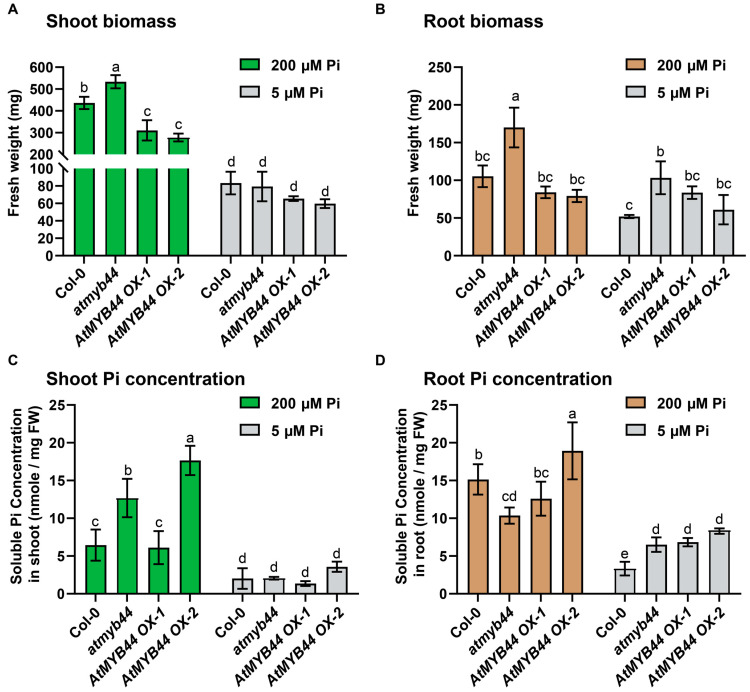
AtMYB44 functions in plant growth under Pi-starvation conditions. (**A**) Shoot and (**B**) root biomass measurements on four-week-old Col-0, *atmyb44*, *AtMYB44 OX-1* and *OX2* lines, grown in a hydroponic culture system with high (200 µM) and low (5 µM) Pi concentrations. Shoot and root biomass was examined as fresh weight for each Arabidopsis plant. Soluble Pi concentration in (**C**) shoot and (**D**) root of four-week-old Col-0, *atmyb44*, *AtMYB44 OX-1* and *OX2* plants grown in a hydroponic culture system with high (200 µM) and low (5 µM) Pi concentrations. The data are presented as mean ± SD (three technical replicates and three technical repeats). The different lowercase letters indicate significant differences in biomass (**A**,**B**) and soluble Pi concentration (**C**,**D**), determined by the Tukey’s test (*p* < 0.05).

**Figure 5 plants-12-03617-f005:**
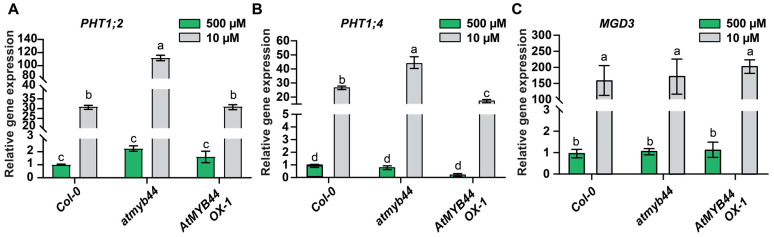
Expression of *PHT1* is enhanced in *atmyb44* roots. Relative expression level of *PHT1;2* (**A**), *PHT1;4* (**B**) and *MGD3* (**C**) in WT (Col-0), *atmyb44* and AtMYB44 OX-1 roots, under Pi-sufficient (500 µM) and Pi-starvation (10 µM) conditions. Arabidopsis seedlings were transplanted onto medium with 500 µM, or 10 µM Pi, 5 days after germination on solid medium with 500 µM Pi. Arabidopsis seedlings were harvested 7 days after Pi-sufficient (500 µM), or Pi-starvation (10 µM) treatment. Arabidopsis *Actin* was used as an internal control to normalize the qRT-PCR results. The data are presented as mean ± SD (three technical replicates and three technical repeats). The different lowercase letters indicate significant differences in relative gene expression levels, determined by Tukey’s test (*p* < 0.05).

**Figure 6 plants-12-03617-f006:**
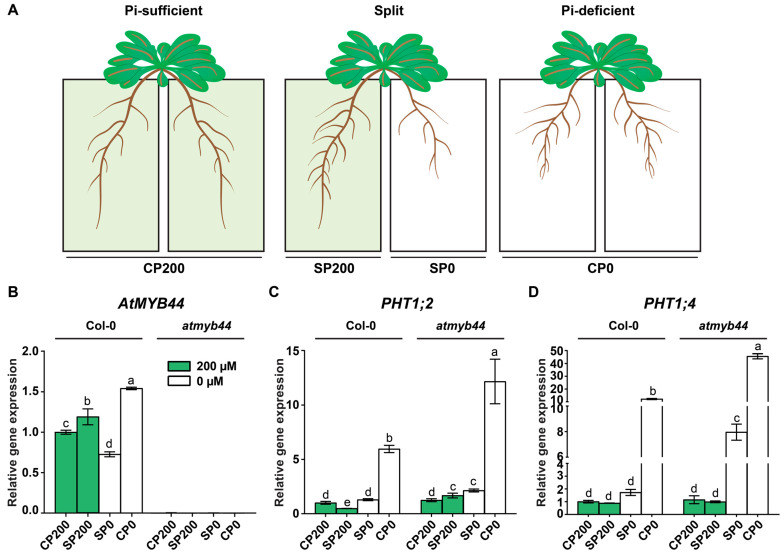
*AtMYB44* is a systemically regulated gene in response to an imposed Pi-starvation stress. Expression of endogenous *AtMYB44* in roots of four-week-old WT (Col-0) and *atmyb44* plants grown in split-root systems. (**A**) Schematic diagram illustrating the experimental design to detect systemic or local signaling under Pi-starvation conditions using a split-root system. CP200 and CP0 indicate homogenous Pi-sufficient (200 µM) and deficient (0 µM) treatment on roots, respectively. SP200 and SP0 indicate the half of the root system on a plant supplied with Pi-sufficient (200 µM) or deficient (0 µM) treatment, respectively. Arabidopsis roots were harvested 7 days after transferring plants into the split-root systems. (**B**) *AtMYB44* expression is systemically regulated. *AtMYB44* expression was examined in WT (Col-0) and *atmyb44*, using a split-root system. (**C**) *PHT1;2* and (**D**) *PHT1;4* expression in WT (Col-0) and *atmyb44* using a split-root system. Arabidopsis *Actin* was used as an internal control to normalize the qRT-PCR results. The data are presented as mean ± SD (three technical replicates and three technical repeats). The different lowercase letters indicate significant differences in relative gene expression levels, determined by Tukey’s test (*p* < 0.05).

**Figure 7 plants-12-03617-f007:**
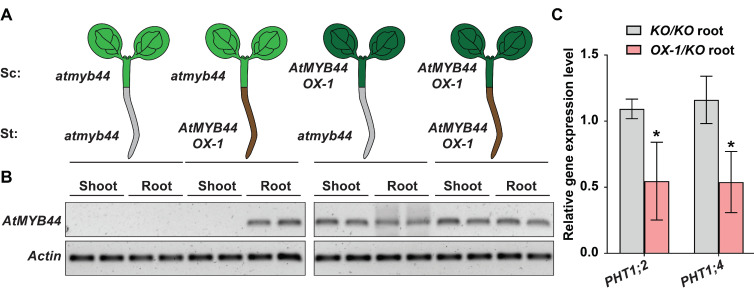
*AtMYB44* plays a role in systemic *PHT1;2* and *PHT1;4* regulation. (**A**) Schematic diagram of Arabidopsis micrografting between *atmyb44* and *AtMYB44 OX-1* seedlings. (**B**) *AtMYB44* is graft-transmissible from an AtMYB44 *OX-1* scion to an *atmyb44* rootstock. (**C**) Expression of *AtPHT1;2* and *AtPHT1;4* is decreased in the heterografted *atmyb44* rootstock. Arabidopsis *Actin* was used as an internal control to normalize the qRT-PCR results. *KO* and *OX-1* indicate *atmyb44* mutant and *AtMYB44* OX-1 line, respectively. The data are presented as mean ± SD (three technical replicates and three technical repeats). Asterisks indicate significantly different values with homografted *atmyb44* plants. (Student’s *t*-test, *p* < 0.05).

**Figure 8 plants-12-03617-f008:**
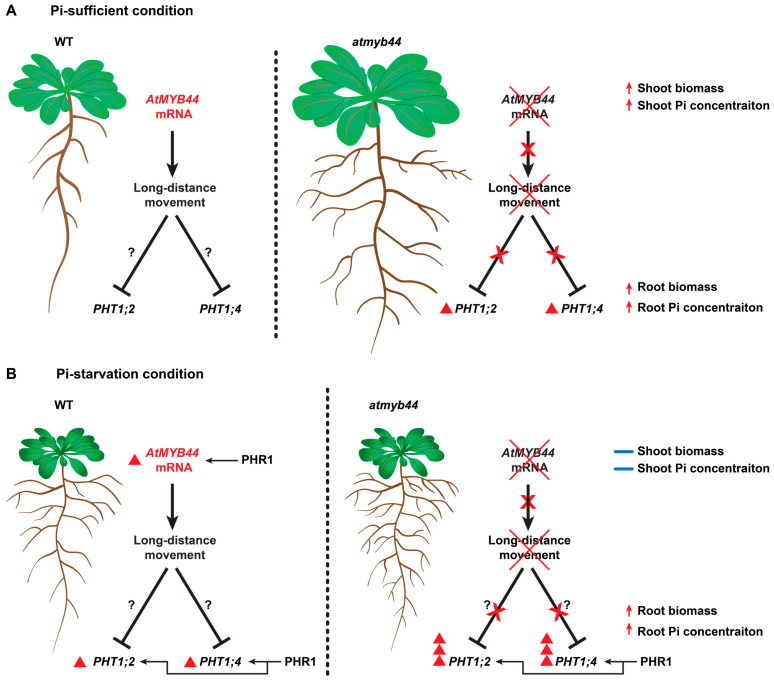
Schematic model of mobile *AtMYB44*-mediated *PHT1;2* and *PHT1;4* regulation in Arabidopsis. (**A**) The mobile *AtMYB44* transcript is long-distantly transported from source leaves to roots via the phloem and serves as a systemic signal to negatively regulate the expression of the Pi transporters, *PHT1;2* and *PHT1;4*, in Arabidopsis roots. In the *atmyb44* mutant, *AtMYB44* mRNA is absent, thereby diminishing inhibition of *PHT1;2* and *PHT1;4* expression in roots. It results in increases of biomass and soluble Pi concentration in both shoots and roots of the *atmyb44* mutant, compared to WT. (**B**) Under Pi-starvation conditions, PHR1 recognizes the promoter region of *AtMYB44* and *PHT1;2/PHT1;4* in shoots and roots, respectively, to induce those expressions. As *AtMYB44* expression is abolished in the *atmyb44* mutant, *PHT1;2* and *PHT1;4* expression is more enhanced in *atmyb44* roots, compared to WT, due to absence of the negative regulatory factor, mobile *AtMYB44* mRNA. Red darts and arrows indicate the increased level of designated gene expression and traits. Blue bars indicate a similar level of shoot biomass and shoot Pi concentration between WT and *atmyb44* under Pi-starvation conditions. Question marks indicate the potential mechanisms that remain to be further investigated in the future. The Red X marks indicate systemic signaling mechanism, mediated by mobile *AtMYB44* mRNA, are not functional in *atmyb44* mutant.

## Data Availability

The data in this study are available within the article.

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
