# Peer review of "Phloem-Mobile MYB44 Negatively Regulates Expression of PHOSPHATE TRANSPORTER 1 in Arabidopsis Roots"

_plants, 2023, doi:10.3390/plants12203617_

Round 1
Reviewer 1 Report
In this MS the authors analyzed and characterized the role of MYB44 on local and systemic responses to phosphate starvation. They found that MYB44 expression was induced in Arabidopsis plants exposed to Pi starvation conditions in shoots and roots. MYB44 mutation triggers the Pi accumulation in roots. Expression analysis revealed that MYB44 could negatively regulate PHT1;2 and PHT1;4 and split-root system and grafting analysis showed that mRNA of MYB44 could be mobilized from shoot to root and regulate systemic responses in Arabidopsis. The MS shows interesting information supported with strong evidence. However, I have comments and observations that the authors should include to be accepted this work.
Major comments:
1.- L96-98. The authors mention that they decided investigate the role of MYB44 because it was induced in shoot and root in response to phosphate starvation. However, I think that authors should have a stronger justification because AtMYB77 shows an interesting expression patter; is repressed in shoot and over expressed in roots; which could acts also a mobile element controlled by phosphate levels.
2.- Fig. S2. The authors analyzed the effect of phosphate starvation on primary root elongation in WT and myb44 mutant, and they found that myb44 did not show a distinct phenotype to WT seedlings. The authors should include MYB44OX1 and MYBOX-2 plants in this study and analyze root elongation.
I could suggest to the authors a second experimental strategy for this experiment to validate the effect of Pi starvation. The authors could use four-old-seedlings WT, myb44, MYB44OX1, MYB44OX4 seedlings grown in 500 uM Pi, then to transfer to growth media with 500 uM Pi (Control) and 10 uM Pi, and analyze every day the root growth.
3.- Fig. 2. The induction of MYB44 expression is evident, but the pictures have very low quality. In results section, the authors mention that MYB44 expression is induced in primary root tip. However, the figure legend indicates that Fig. 2E, F shows lateral roots. The authors should verify it.
I suggest that authors should analyze the MYB44 expression every 24 hrs after exposition to Pi starvation conditions. This could indicate if MYB44 is an early-responsive Pi starvation element.
The authors should realize an analysis more specific of MYB44 expression in primary root tip. For this, the authors should to take pictures of distinct microscopy planes (e.g. QC tissues and epidermal tissue). This experiment could be analyzed by confocal microcopy in seedlings stained with propidium iodide, since the transgenic line contains also GFP reporter. This information will indicate the patter expression of MYB44.
4.- Fig. 3. Here authors used 200 uM as high Pi-sufficient, whereas in the others experiment used 500 uM, why?. They should justify it.
5.- Fig. 4. Authors analyze the Pi contentment and they found that myb44 mutant accumulates more Pi than WT seedlings. However, MYB44OX1 shows Pi contents similar than WT, but MYBOX4 showed the highest level. The authors should discuss it, and they should include the genotypification of MYBOX1 and MYBOX4 lines (fold change of MYB44 expression).
6.- Because MYB44, 70, 73, and 77 could have functional redundance, I consider important that authors generate double or triple mutants for MYB44, 70, 73, 77 and analyze it in Pi starvation.
7.- It could be interesting if authors analyze the effect of Pi starvation in myb70, myb73, and myb77 single mutants.
8.- The authors could analyze (could be by RT-qPCR) another Pi starvation genes in WT and myb44 mutant (e.g. AtPT1 and AtPT1, two phosphate transporter, one with constitutive expression and the second with inducible expression).
9.- The authors could include a second line for MYB44 mutation differed to the used in this work.
Author Response
Reviewer 1 comments:
In this MS the authors analyzed and characterized the role of MYB44 on local and systemic responses to phosphate starvation. They found that MYB44 expression was induced in Arabidopsis plants exposed to Pi starvation conditions in shoots and roots. MYB44 mutation triggers the Pi accumulation in roots. Expression analysis revealed that MYB44 could negatively regulate PHT1;2 and PHT1;4 and split-root system and grafting analysis showed that mRNA of MYB44 could be mobilized from shoot to root and regulate systemic responses in Arabidopsis. The MS shows interesting information supported with strong evidence. However, I have comments and observations that the authors should include to be accepted this work.
Major comments:
1.- L96-98. The authors mention that they decided investigate the role of MYB44 because it was induced in shoot and root in response to phosphate starvation. However, I think that authors should have a stronger justification because AtMYB77 shows an interesting expression pattern; is repressed in shoot and over expressed in roots; which could act also a mobile element controlled by phosphate levels.
- Thank you for this comment. In the previous report, published by Thieme et al., [40], AtMYB44 and 70, not 77, were identified as mobile mRNAs and Pi-stress seems to lead to the mobility of AtMYB44 mRNA. We added this information in the revised manuscript at line 124-128.
2.- Fig. S2. The authors analyzed the effect of phosphate starvation on primary root elongation in WT and myb44 mutant, and they found that myb44 did not show a distinct phenotype to WT seedlings. The authors should include MYB44OX1 and MYBOX-2 plants in this study and analyze root elongation.
I could suggest to the authors a second experimental strategy for this experiment to validate the effect of Pi starvation. The authors could use four-old-seedlings WT, myb44, MYB44OX1, MYB44OX4 seedlings grown in 500 uM Pi, then to transfer to growth media with 500 uM Pi (Control) and 10 uM Pi, and analyze every day the root growth.
- We performed the root phenotyping with WT, myb44 mutant, MYB44 OX-1 and MYB44 OX-2 lines and prepared new Figure S3 in the manuscript. Consistent with the previous report [25,41] (we used the same OX lines which were generated in the previous study), OX lines showed shorter primary root (PR) length, compared to WT and myb44 mutant under Pi-sufficient conditions. However, under Pi-starvation conditions, PR growth performance was quite similar between WT, atmyc44 mutant, MYB44 OX-1 and MYB44 OX-2. In addition, we also add the information of lateral root number in MYB44 OX-1 and MYB44 OX-2 lines in this Figure S3. We described this new result at line 161-168 in the revised manuscript.
3.- Fig. 2. The induction of MYB44 expression is evident, but the pictures have very low quality. In results section, the authors mention that MYB44 expression is induced in primary root tip. However, the figure legend indicates that Fig. 2E, F shows lateral roots. The authors should verify it.
I suggest that authors should analyze the MYB44 expression every 24 hrs after exposition to Pi starvation conditions. This could indicate if MYB44 is an early-responsive Pi starvation element.
The authors should realize an analysis more specific of MYB44 expression in primary root tip. For this, the authors should to take pictures of distinct microscopy planes (e.g. QC tissues and epidermal tissue). This experiment could be analyzed by confocal microcopy in seedlings stained with propidium iodide, since the transgenic line contains also GFP reporter. This information will indicate the patter expression of MYB44.
- We add the results of GUS staining at the primary root tip as new Figure 2C and 2D, which took images from the same specimens in Figure 2A and 2B. We compared the GUS expression at the primary root tip under Pi-sufficient (Figure 2C) and starvation (Figure 2D) conditions, in order to show the clear AtMYB44 expression in response to Pi-stress at the primary root tip.
- in Figure 2G and 2H (the previous Figure 2E and 2F), we showed different patterns of AtMYB44 expression in mature lateral roots. We added description of this result in the revised manuscript at line 138-141. And we also add the image of GUS staining at the primary root tip in the inset of Figure 2B.
- We added the new Figure 2K-2L and Figure S2 to show where the AtMYB44 is expressed in the primary root tip, using the confocal microscopy. We detected AtMYB44 expression in endodermis, cortex layer and epidermis under Pi-stress conditions. We described this information at line 141-143 in the revised manuscript.
- Based on the editor’s comments, we also performed the time-course experiments to profile the AtMYB44 expression in shoots and roots (Figure 2M-2N) and showed increased expression level of AtMYB44 in response to Pi-starvation stress. We described this information at line 143-145 in the revised manuscript.
4.- Fig. 3. Here authors used 200 uM as high Pi-sufficient, whereas in the others experiment used 500 uM, why?. They should justify it.
- In this study, we used three systems to test plant response under Pi-stress conditions: solid medium, hydroponics and sand culture. We used 500 uM to perform this physiological test on solid medium and in sands and 200 uM in hydroponics as Pi-sufficient conditions. As we used solid medium for the experiments in Figure 3, it should be 500 uM as Pi-sufficient conditions. We apologize for our typo in figure 3. In the revised manuscript, we corrected figures to clearly indicate what concentration we used for each experiment.
5.- Fig. 4. Authors analyze the Pi contentment and they found that myb44 mutant accumulates more Pi than WT seedlings. However, MYB44OX1 shows Pi contents similar than WT, but MYBOX4 showed the highest level. The authors should discuss it, and they should include the genotypification of MYBOX1 and MYBOX4 lines (fold change of MYB44 expression).
- In this study, we used MYB44 OX-1 and 2 which were provided from other research group [reference 25,41]. They confirmed the expression level of these OX lines in their articles. We cannot clearly explain the result of Pi content in MYB44 OX-1 and 2, compared with WT and mutant in this study. But we propose the possible reason in the main text from line 305-311.
6.- Because MYB44, 70, 73, and 77 could have functional redundance, I consider important that authors generate double or triple mutants for MYB44, 70, 73, 77 and analyze it in Pi starvation.
- Thank you for providing this great idea to us. In fact, one of our ongoing projects is to generate double or triple mutants among MYB44, 70,73 and 77 to test the role of those MYBs in Pi-starvation systemic signaling pathway.
7.- It could be interesting if authors analyze the effect of Pi starvation in myb70, myb73, and myb77 single mutants.
- Similar our response to the question 6, myb70, myb73, and myb77 are using in other projects of our lab to examine each MYB function in Pi homeostasis.
8.- The authors could analyze (could be by RT-qPCR) another Pi starvation genes in WT and myb44 mutant (e.g. AtPT1 and AtPT1, two phosphate transporter, one with constitutive expression and the second with inducible expression).
- In Figure 5, we add the RT-qPCR result to analyze the expression of type B monogalactosyl diacylglycerol synthase 3 (MGD3), which is one of Pi starvation response genes [51]. As MGD3 was not predicted as AtMYB44 target (Table S2), we analyzed expression of MGD3 in myb44 mutant and AtMYB44 OX-1 line to compare that of PHT1;2 and PHT1;4. As shown in Figure 5C, the expression pattern of MGD3 in WT (Col-0) was similar with that in atmyb44 mutant and AtMYB44 OX-1 line, suggesting that AtMGD3 might not be the AtMYB44 target. We added this information at line 204-209 in the manuscript.
9.- The authors could include a second line for MYB44 mutation differed to the used in this work.
- We understand this suggestion is great and will strengthen our hypothesis regarding MYB44 function in Pi-stress systemic signaling. The reason why we chose this mutant line in our study is that many research groups confirmed the knock-out of MYB44 expression in this line and have used the same mutant for various studies, rather than other mutant lines [reference 24,25,26,27,41].
Reviewer 2 Report
This paper entitled “Phloem-mobile MYB44 negatively regulates expression of PHOS-2 PHATE TRANSPORTER 1 in Arabidopsis roots”was conducted to explore AtMYB44 as a phloem-mobile mRNA, an Arabidopsis homolog of Cucumis sativus MYB44, that is responsive to the Pi-starvation stress.
1. Figure S1, 132 Arabidopisis MYB family members were analyzed, how about MYB family in Cucumis sativus ? How homology of those members in Cucumis sativus? If MYB44 (CsMYB44, Csa6G491690) exhibited a systemic mobility as described in Zhang et al., 2016, why other members with high homology were not mobile? What’s the potential differences between them such as motif difference?
2. Figure S2A, which ones are WT and atmyb44?
3. Figure 7, better to test the mobility of AtMYB44 in different Pi condition (Pi-sufficient (200 μM) or deficient (0 μM) treatment, so as to understand the quantitative movement difference of AtMYB44.
No comments
Author Response
This paper entitled “Phloem-mobile MYB44 negatively regulates expression of PHOS-2 PHATE TRANSPORTER 1 in Arabidopsis roots”was conducted to explore AtMYB44 as a phloem-mobile mRNA, an Arabidopsis homolog of Cucumis sativus MYB44, that is responsive to the Pi-starvation stress.
- Figure S1, 132 Arabidopisis MYB family members were analyzed, how about MYB family in Cucumis sativus ? How homology of those members in Cucumis sativus? If MYB44 (CsMYB44, Csa6G491690) exhibited a systemic mobility as described in Zhang et al., 2016, why other members with high homology were not mobile? What’s the potential differences between them such as motif difference?
- The recent study reported identification of 112 MYB family members in Cucumis sativus [30]. Interestingly, based on the previous report (Zhang et al., 2016), CsMYB44 (Csa6G491690) was identified as the only Pi-stress responsive mobile mRNA, encoded for the MYB transcription factor.
The current cucumber database mostly does not include the sequence information of complete UTR regions; therefore, it is a challenge to analyze characterized motifs which could contribute to phloem mRNA mobility, e.g. modified base 5-methylcytosine (m5C), CU- or tRNA-like motifs, in cucumber.
We described this information in the main text at line 91-97 and added new phylogenetic tree in Figure S1A (the previous Figure S1A becomes Figure S1B).
- Figure S2A, which ones are WT and atmyb44?
- Thank you for pointing out this. We indicated WT, atmyb44 as well as OX lines in Figure S2A.
- Figure 7, better to test the mobility of AtMYB44 in different Pi condition (Pi-sufficient (200 μM) or deficient (0 μM) treatment, so as to understand the quantitative movement difference of AtMYB44.
- Thank you for this comment and it is a great suggestion. We emphasized the role of mobile MYB44 in PHT1 regulation in roots. As the expression level of MYB44 was increased in shoots under Pi-starvation conditions (Figure 2M) and the mobility of AtMYb44 mRNA (Figure 7) was detected in the heterografting assay, it is plausible that higher rate of AtMYB44 mobility might be detected in roots under Pi-starvation conditions, compared to Pi-sufficient conditions. However, we cannot exclude the possibility that other phloem factors, e.g. RNA-binding proteins, regulates AtMYB44 mobility in response to Pi-starvation stress. As the different evaluation regarding the mobile mechanism of AtMYB44 mRNA is necessary to address this question, we hope to deliver this information in the next manuscript.
Reviewer 3 Report
1. Has the MYB gene been reported to respond to phosphorus deficiency conditions? I think a description of relevant studies should be added to the introduction section. Also, a paragraph describing the overall conclusion of the article should be added at the end of the introduction section to make the article more complete.
2. Lines 35-38, please add references.
3. In Figure S1, the addition of bootstrap values is necessary. In addition, what is the homology ratio of MYB44, MYB70, MYB73, and MYB77 in Arabidopsis? What do the sequence comparison results look like?
4. In Figure S2, what is the sufficient concentration of phosphorus (200 μm or 500 μm)? Which plants are WT or atmyb44 in Figure S2B? Significance analyses should be added to Figure S2B.
5. In the hydroponic treatment, why did the concentration of phosphorus deficiency become 5 μm? Surprisingly, there was a significant increase in phosphorus content in AtMYB OX-2 shoots (Fig.4C). In addition, after 4 weeks of treatment, phosphorus content in atmyb44 roots decreased, but phosphorus content in shoots increased, which seems to be inconsistent with common sense. And there was a significant increase in phosphorus content in roots of both atmyb44 and AtMYB44 OX lines after phosphorus deficiency treatment.
6. Lines 169-180, it would be helpful to know how much higher or lowered, whether the increase was significant.
7. In the Results 2.5 section, the authors only showed the binding relationship between AtMYB44 and PHT by analysis, which I think should be verified by adding relevant experiments.
8. At line 202, "We did not detect any significant differences in expression levels of PHT1;2 and PHT1;4 in AtMYB44 OX-1 ... ...", I do not think so, apparently PHT1;4 was significantly downregulated in AtMYB44 OX-1 according to Figure 5B.
9. In Figure 6, I think the addition of a schematic of the split-root test is a bit more intuitive for understanding the design of the assays. The text is not clear about the split-root test, and I suggest that it be revised. In addition, is it possible to show that MYB44 is a systemic regulator just by measuring the expression of two genes? In Col-0, it seems that MYB and PHT have similar expression patterns, which is contrary to the conclusion that MYB negatively regulates PHT, what is the reason for this?
10. Except for MYB44, are the other three genes (MYB70, MYB73 and MYB77) able to regulate PHT gene expression? This point seems to need to be elaborated to account for the importance of MYB44.
11. Although the article demonstrates the mobility of MYB44 mRNA, the authors did not use grafted plants during the verification of function. Since ungrafted plants contain native MYB44 mRNA and MYB44 in both leaves and roots responds to phosphorus deficiency, the experimental design and logic of this article does not make sense and does not support the conclusion in the title.
12. Lines 236, 241, 459, please make a harmonized statement. c200/c0 or cp200/cp0?
13. Two methods of data processing are used in the MS (student's test and turkey's test), please harmonize them.
Moderate changes to the language of the article are recommended.
Author Response
Reviewer 3 comments:
- Has the MYB gene been reported to respond to phosphorus deficiency conditions? I think a description of relevant studies should be added to the introduction section. Also, a paragraph describing the overall conclusion of the article should be added at the end of the introduction section to make the article more complete.
- We add a paragraph to introduce the earlier reports of MYB function in Pi-stress signaling at the Introduction section (Line 61-76). And we also add one paragraph to summarize our study at Introduction section (Line 76-81).
- Lines 35-38, please add references.
- We add references for the description at line 35-38.
- In Figure S1, the addition of bootstrap values is necessary. In addition, what is the homology ratio of MYB44, MYB70, MYB73, and MYB77in Arabidopsis? What do the sequence comparison results look like?
- We added new Figure S1A with the bootstrap value, to show the AtMYB44 and CsMYB44 clades together in the phylogenetic tree.
- We also added the homology information of AtMYBs at line 104-105 in the revised manuscript.
- The sequence alignment of MYB44, MYB70, MYB73, and MYB77 was already reported [26], therefore, we don’t include this information in our manuscript.
- In Figure S2, what is the sufficient concentration of phosphorus (200 μm or 500 μm)? Which plants are WT or atmyb44in Figure S2B? Significance analyses should be added to Figure S2B.
- Many thanks for pointing out this issue. This study used 3 systems to treat Pi-stress in plants; hydroponics, solid media and sand culture system. For the Pi-sufficient treatment, we used 200 uM for hydroponics and 500 uM for other systems. We used 500 uM for Figure S3 and It was a typo in the figure legend. We apologize for our typo in figure S3 and carefully revised our manuscript to clarify what concentration we used for experiments.
- During the revision process, we prepared new Figure S3 in which WT and atmyb44 as well as AtMYB44 OX-1 and AtMYB44 OX-2 lines showed the root growth performance under Pi-sufficient and starvation conditions. And we also added the statistical information for this figure.
- In the hydroponic treatment, why did the concentration of phosphorus deficiency become 5 μm? Surprisingly, there was a significant increase in phosphorus content in AtMYBOX-2 shoots (Fig.4C). In addition, after 4 weeks of treatment, phosphorus content in atmyb44 roots decreased, but phosphorus content in shoots increased, which seems to be inconsistent with common sense. And there was a significant increase in phosphorus content in roots of both atmyb44and AtMYB44 OX lines after phosphorus deficiency treatment.
- As the nutrients can be directly fed to the plant root system in hydroponics, the hydroponic system is a more efficient method to control the mineral nutrient levels which plant biologists test the plant responses to nutrient starvation stresses, compared to soil and solid media. Thus, if we used the same nutrient concentration for the hydroponics, which is used for Arabidopsis growth in soil and solid media, it is possible that Pi 10 μM could not be enough for Pi-starvation stress on Arabidopsis in hydroponics. We added a simple note at line 173-176.
- As the reviewer pointed out, we are also aware of unexpected phenotypes in AtMYB44 OX lines. Our current study cannot address this question clearly. However, as MYB44 can contribute to multiple signal transduction pathways, ectopic MYB44 expression might result in unexpected signaling under Pi-stress conditions. We mentioned this possibility at line 304-311 in the discussion section.
- Lines 169-180, it would be helpful to know how much higher or lowered, whether the increase was significant.
- We added the information of fold difference in the biomass and Pi concentration at line 177-189.
- In the Results 2.5 section, the authors only showed the binding relationship between AtMYB44 and PHT by analysis, which I think should be verified by adding relevant experiments.
- Based on the qRT-PCR and in silico analysis, we suggested that AtMYB44 targets the promoter of PHT1;2 and PHT1;4, where it acts as a negative regulator. The ConnecTF, we used for in-silico analysis to analyze the target sites of AtMYB44, is a platform which consists of the information regarding interaction of TF-target genes, which has been validated using various approaches, e.g. ChIP/DAP-seq, aetc. Therefore, the binding of AtMYB44 on the promoter region of PHT1;2 and PHT1;4 was tested in the previous literature and this information was used to develop ConnecTF. We agree with both the reviewer and editor’s opinion on this comment, and revise this point with less definitive description and as future works at the line 208-210 and line 322-324 in the revised manuscript.
- At line 202, "We did not detect any significant differences in expression levels of PHT1;2and PHT1;4in AtMYB44 OX-1 ... ...", I do not think so, apparently PHT1;4 was significantly downregulated in AtMYB44 OX-1 according to Figure 5B.
- We appreciate you giving us this comment. We revised this point at line 203.
- In Figure 6, I think the addition of a schematic of the split-root test is a bit more intuitive for understanding the design of the assays. The text is not clear about the split-root test, and I suggest that it be revised. In addition, is it possible to show that MYB44is a systemic regulator just by measuring the expression of two genes? In Col-0, it seems that MYB and PHT have similar expression patterns, which is contrary to the conclusion that MYB negatively regulates PHT, what is the reason for this?
- Based on the reviewer’s comment, we added an experimental scheme at Figure 6A, to make clear about the spilt-root assay.
- In this study, we proposed that MYB44 functions in systemic regulation of PHT1;2 and PHT1;4 expression. Thibaud et al. [12] profiled the genes which could be systemically or locally regulated in response to Pi-starvation stress and PHT1 was one of listed systemically-regulated genes. We agree that it would be better to profile the expression pattern of other genes to test the role of mobile MYB44 mRNA in Pi-starvation stress-signaling network, using high-throughput assays (e.g. RNA-seq). However, it will be our future study and we hope we can provide this information in the following publication.
- We discussed the point regarding the similar expression pattern of MYB44 and PHT1 in WT under Pi-starvation condition, at line 363-375 in the manuscript. Even though we cannot clearly address this question in our current study, we presume that it might be a fine-turning regulation, mediated by MYB44 for negative control over PHT1 expression under Pi-starvation conditions, in order to balance between Pi uptake and other adaptive plant responses.
- Except for MYB44, are the other three genes (MYB70, MYB73and MYB77) able to regulate PHT gene expression? This point seems to need to be elaborated to account for the importance of MYB44.
- Thank you for providing this question to us. In the previous report, published by Thieme et al., [40], AtMYB44 and 70, not 77, were identified as mobile mRNAs and Pi-stress seems to lead to the mobility of AtMYB44 mRNA. We describe this point at line 124-128 in the revised manuscript, to provide additional justification regarding why we decided to perform the further study with AtMYB44. As this manuscript deals with the role of AtMYB44 in Pi-stress signaling and characterization of other MYB members is in our other ongoing projects, we hope providing this result in other publication.
- Although the article demonstrates the mobility of MYB44 mRNA, the authors did not use grafted plants during the verification of function. Since ungrafted plants contain native MYB44mRNA and MYB44in both leaves and roots responds to phosphorus deficiency, the experimental design and logic of this article does not make sense and does not support the conclusion in the title.
- We performed the micrografting between MYB44 OX-1 and atmyb44 mutant, to test whether AtMYB44 is detected in heterografted atmyb44 mutant.
- Lines 236, 241, 459, please make a harmonized statement. c200/c0 or cp200/cp0?
- Thank you for your comments. We decided to use “CP200” and “CP0” to describe Pi concentration of nutrient solution for the split-root system and revised this point in our manuscript.
- Two methods of data processing are used in the MS (student's test and turkey's test), please harmonize them.
- We reanalyzed our data, using the Turkey’s test method, to obtain the statistical information in the revised manuscript. As we compared the data with only control to examine the significant difference in Figure 1C-D and 7C, we used the student t-test for those data.
Round 2
Reviewer 1 Report
The authors performed several experiments to validate the main conclusions and they answered all my suggestions. I don't have further comments.